# Proof-of-Principle of a Cherenkov-Tag Detector Prototype

**DOI:** 10.3390/s20123437

**Published:** 2020-06-18

**Authors:** Giuseppe Gallo, Domenico Lo Presti, Danilo Luigi Bonanno, Giovanni Bonanno, Paola La Rocca, Santo Reito, Francesco Riggi, Giuseppe Romeo

**Affiliations:** 1National Institute for Astrophysics (INAF), Catania Astrophysical Observatory, 95123 Catania, Italy; giovanni.bonanno@inaf.it (G.B.); giuseppe.romeo@inaf.it (G.R.); 2Department of Physics and Astronomy “E. Majorana”, University of Catania, 95123 Catania, Italy; domenico.lopresti@unict.it (D.L.P.); danilo.bonanno@ct.infn.it (D.L.B.); paola.larocca@ct.infn.it (P.L.R.); francesco.riggi@ct.infn.it (F.R.); 3National Institute for Nuclear Physics (INFN), Catania Division, 95123 Catania, Italy; santo.reito@ct.infn.it

**Keywords:** muography, Cherenkov radiation, silicon photo-multiplier, VMM3a chip, particle detectors

## Abstract

In a recent paper, the authors discussed the feasibility study of an innovative technique based on the directionality of Cherenkov light produced in a transparent material to improve the signal to noise ratio in muon imaging applications. In particular, the method was proposed to help in the correct identification of incoming muons direction. After the first study by means of Monte Carlo simulations with Geant4, the first reduced scale prototype of such a detector was built and tested at the Department of Physics and Astronomy "E. Majorana" of the University of Catania (Italy). The characterization technique is based on muon tracking by means of the prototype in coincidence with two scintillating tiles. The results of this preliminary test confirm the validity of the technique and stressed the importance to enhance the Cherenkov photons production to get a signal well distinguishable with respect to sensors and electronic noise.

## 1. Introduction

Today we know that cosmic-ray muons are continuously irradiating everything on the Earth’s surface. At sea level, the integral intensity of vertical muons is approximately 70 $m^{-2}s^{-1}{\mathrm{sr}}^{-1}$, with a rough $\cos^{2}\left(\theta \right)$ dependence, where is the zenith angle measured from the vertical axis, which is characteristic for muons with energy nearly equal to 3 GeV [1]. The fundamental properties of muons, in particular their mass and lifetime, explain why they weakly interact with matter, and, hence, have a high penetration power, and dominate by more than one order of magnitude the flux of cosmic particles at sea level [2,3]. In other words, muons are the best candidate to inspect the inner structure of very large objects [4]. This kind of radiography by means of cosmic muons is usually called “muography”.

The idea of using muons as a non-invasive probe dates back to 1955 with the pioneer work of E.P. George [5], even if the first historic application of muography which had big resonance is the famous experiment of L. Alvarez in 1970 [6] with spark chambers inside the Chephren’s pyramid in Egypt. Due to the historical importance of the target, almost fifty years later, in 2017, the discovery by means of muon radiography of a secret chamber inside the 4500 years old Great Pyramid in Giza, Egypt, came to the fore with the same clamor [7]. However, during the last fifteen years, the applications of muon-radiography to several fields, such as geology, mining and civil engineering, are continuously expanding. In particular, the first attempts of studying the inner structure of a volcano using muons belong to K. Nagamine in 1995 [8] and H. Tanaka in 2003 [9]. Since that moment, many successful applications of muon imaging by the Japanese group followed [10,11,12] and also by European groups [13,14,15,16,17,18,19]. For a more detailed description of cosmic ray muons as imaging tools for multidisciplinary applications, interested readers are referred to [20,21].

Each kind of muon imaging application requires a tracker detector, usually referred to as "muon telescope", able to track particles which traverse it coming from two sides, front and back, one of which is pointed towards the object to be inspected. Starting from the assumption that atmospheric muon flux is downward oriented only, the incoming direction of muons inside the field of view of a telescope can be distinguished from the slope of reconstructed trajectories. In a previous paper [22], we proposed a new kind of Cherenkov detector, as a part of the R&D activities of the project for Muography of Etna Volcano (MEV) [23,24]. The feasibility study, developed by means of Monte Carlo simulations with Geant4, explored the usage of such detector to reduce the background noise due to "upward-scattered" muons, i.e., secondary cosmic-ray particles scattered near the detector that traverse it perfectly miming the tracks of downward going particles in opposite directions [14]. This kind of noise bothers all muography experiments dedicated to the investigation of the inner structure of a volcano and, in particular, the cases in which the detector which tracks the particles is horizontally oriented, like the first muon tracker built by the MEV collaboration [25]. The principal solution proposed to identify and reduce these background noise tracks, is the time-of-flight (TOF) measurement of the particles in traversing the detector [26]. However, the time resolution of this technique, typically around 1 ns, and the limited size of muon telescopes, usually within 1–2 m, do not allow to completely remove the uncertainty about the incoming direction of fast muons.

Just for a brief reminder, the working principle of the new Cherenkov-tag detector we propose is based on the directionality of the radiation emitted when a charged particle with sufficient energy passes through a transparent material. The design of the detector is based on two tiles of transparent material, e.g., Plexiglass, placed side by side on their larger surface and divided by a light absorbing coating foil, which also covers the lateral faces. The opposite face of each tile is equipped with an array of silicon photo-multipliers (SiPMs), with a well-suited spatial distribution. The goal is to ensure that when a charged particle traverses both tiles, the Cherenkov radiation produced will be detected only by the SiPMs of the exit tile, while the photons emitted in the first radiator are blocked by the light absorbing coating. In order to make this clearer, in Figure 1 we report an example from [22], in which a muon (red track), with kinetic energy equal to $10^{2}$
GeV, traverses the detector from up to down and produces Cherenkov photons (green lines). In the first tile traversed, the photons stop in light absorbing foil between the two radiators; in the second tile traversed, instead, the photons are directed toward the instrumented face and can be detected by the SiPMs.

In this work, we present the results of the first experimental test for validating the technique of the new Cherenkov-tag detector. For this purpose, a reduced scale prototype was built with the means available in our laboratory at the Department of Physics and Astronomy “E. Majorana” of the University of Catania (Catania, Italy). The test consists in measuring the flux of nearly vertical cosmic-ray secondary muons with the prototype and comparing the results with a reference detector consisting of two scintillating tiles, vertically arranged above the Cherenkov prototype. In order to take into account the different sensitive areas of the two measurement systems, a geometric simulation was performed and the comparison between the two measured fluxes is in accordance with what is expected. This confirms the validity of the working principle of the Cherenkov-tag detector, with a high efficiency as expected from the feasibility study.

## 2. Materials and Methods

### 2.1. Cherenkov-Tag Prototype Design and Construction

Figure 2 shows the Cherenkov-tag prototype built for the starting test here discussed. A board with an array of 4×4 SiPMs Hamamatsu S13360-6050PE (6×6${mm}^{2}$ nominal surface area) is installed on the bottom of a 3D printed PLA plastic box custom designed for the test and the lateral inner faces of the box are covered by sheets of light-absorbing material [27]. The size of the board is 10×10${cm}^{2}$ and the regular array of SiPMs, that are 1.5
cm center-to-center equally spaced, is centered on the board. The margins around the SiPM array, required by design constraints, are clearly insensitive, so it is possible to consider an active area of 6×6${cm}^{2}$. The box, after being prepared as just described, was filled with a layer of optical gel, for a thickness of about 2 cm above the board (see Figure 2a). The optical gel used is the two-component A/B SilGel 612 WACKER [28]. This silicone-based medium has a refractive index equal to 1.404 and, thanks to its optical properties, allows the almost undisturbed passage of Cherenkov photons up to the photo-sensors, improving the transmission of UV Cherenkov light with respect to the Plexiglass chosen for the simulations. The two components, with a 1:1 ratio, were mixed with a mechanical stirrer before the box filling. In order to improve the light transmission, the box filled with the gel was placed in a vacuum system, at a pressure equal to 60kPa, to perform a degassing procedure and remove the air bubbles trapped in the gel mixture. After the gel dried completely, the box was closed with a lid, whose inner face is also covered with the same light absorbing material, as shown in Figure 2b. For this starting test, we decided to build a detector with a single layer of radiator and split a measurement in two steps. In the first configuration, the detector is placed horizontally with the sensitive surface of the SiPM upward directed, in order to measure the Cherenkov radiation produced by muons coming from the sky; in the second step, instead, the detector is flipped in the up-down direction. In this latter configuration, it is expected that no Cherenkov photons will be directed toward the SiPM because cosmic-ray muons do not come from the bottom. On the other hand, if a certain number of events is registered also in this configuration, it can be assumed as a measure of the fake events rate to subtract from the rate measured in the up configuration. A sketch of the two measurement configurations is shown in Figure 3. It is as if the detector shown in Figure 1 had been divided into two separate blocks. The areas covered by light diagonal lines pattern represent the insensitive optical gel volume.

### 2.2. SiPMs Front-End Electronics

Front-End (FE) electronics for SiPM signal acquisition consists of a single custom designed board, shown in Figure 4, based on the VMM chip (version 3a), which was developed by Brookhaven National Laboratory (BNL) for the ATLAS experiment at CERN. The VMM chip has 64 channels providing the peak amplitude and the time with respect to an external clock or a trigger signal in a data driven mode. This task can be accomplished thanks to a fast comparator with an individually adjustable threshold, with which each channel is equipped. The chip generates a stream of 38 bits for each event in each interested VMM channel. The first two bits represent a readout flag, next 6 bits account for the channel address, followed by 10 bits associated with the charge collected while the last 20 bits are associated with timing information. Each channel has a de-randomizing FIFO able to store data up to 4 events. Further details about FE electronics can be found in [29].

### 2.3. Test Setup and Read-Out Electronics

In order to select only a portion of the secondary cosmic muons around the vertical direction and measure the efficiency of the Cherenkov-tag prototype with respect to a reference detector, the test setup consists of two scintillating tiles, each coupled to a Photo-Multiplier tube (PMT), by means of a light guide, to read-out the scintillating light generated by the passage of a charged particle. A sketch of the experimental setup is shown Figure 5. The area of both scintillators is 12×12${cm}^{2}$ and they are aligned and placed one over the other 60 cm away. The solid angle subtended by the two scintillators in this geometry is nearly equal to 0.1539
sr.

Both the PMTs and the VMM FE board are connected to a unique Read-Out (RO) board specifically designed to interface with VMM-board and it is based on the System On Module (SOM) manufactured by National Instruments [30]. The SOM, already used for the RO of the MEV muon telescope, is a very compact, flexible, high-performance, deployment-ready embedded computer. The main features are a 667 MHz Dual-Core ARM Cortex-A9 processor, NI Linux Real-Time OS and an Artix-7 programmable Xilinx FPGA. Further details about a SOM based RO can be found in [31].

The presence of a suitable analog output of each PMT, in order to be converted into a LVCMOS signal (the acronym stands for Low Voltage Complementary Metal Oxide Semiconductor and the signal level is equal to 3.3V) compatible with SOM inputs, is passed to a mini-board for amplification, filtering and digitization of the signal. It consists of an amplifier MAX4412, followed by a fast comparator MAX961, with TTL/CMOS compatible output, and a monostable, that maintains the signal at high level for about 200 ns. In this manner the two logic signals corresponding to the two PMTs are sent directly to the SOM, which performs time coincidence. According to the acquisition strategy established for the test, the SOM continuously quereis the VMM memories, but it saves data only when receiving a trigger signal, which in our case is equal to the coincidence of both PMTs, given by the result of the logic AND operation. Data are stored on a local PC wired connected with the SOM. The SOM manages all the VMM settings and a UI developed on LabVIEW allows users to control and monitor them from the PC. When the SOM receives data from VMM, it appends a 26 bits timing information at the 38 bits string coming from VMM chip. This information is related to the clock cycle when the external trigger arrives and it allows to check the coincidence between SiPM’s and scintillators’ events. Data analysis was done by means of dedicated MATLAB code which performs the decoding of VMM binary streams. A preliminary calibration of the VMM response was performed, with prototype in down configuration, in order to find the best SD values for each channel that produce an equalized dark current rate response of the 16 sensors.

## 3. Results

The main parameters of the VMM chip that can be set for the acquisition are gain, peak time and threshold. Gain and peak time were set to their minimum values, respectively 0.5 and 25 ns in order to work properly with SiPM signals and avoid saturation and event pile-up. After some preliminary tests, these considerations were confirmed also for the measurement of Cherenkov light pulses produced in the prototype in study. Therefore, the main parameter which was changed during the tests here discussed is the (voltage) threshold. The VMM chip allows to set a common threshold for all channels (SDT) against which the threshold of each single channel (SD) can be finely adjusted within a range of possible values ranging from 0 to −29 mV with respect to the SDT. A preliminary calibration of the VMM response was performed, with prototype in down configuration, in order to find the best SD value for each channel that resulted in an equalized dark current rate response of the 16 sensors.

Table 1 shows the SDT values fixed for each couple of measurements—one in configuration up and the other in configuration down, together with the respective acquisition times. SDT is the unique VMM parameter changed from one acquisition to another.

The first value we are interested in for each acquisition is the number of SiPM hit by the Cherenkov radiation produced by the passage of a particle through the prototype, that is ensured by the coincidence with both scintillators. During data analysis, it is possible to set a (charge) threshold in order to consider a SiPM hit, because the VMM data contains information about the charge related to how many photons strike on each sensor. Figure 6 shows an example of bar plot to describe the effect of the charge threshold. Gray bars represent the probability that a certain number *n* of SiPM are stricken when there is coincidence with both scintillators; each red bar corresponds to the percentage of corresponding gray bars in which at least one SiPM has a signal greater than the charge threshold. This explanation is useful to introduce the definition of “event correctly tagged”: for the measurement in UP configuration, a coincidence event between both scintillators and the Cherenkov prototype is considered as “correctly tagged” (i.e., the muon’s incoming direction is correctly reconstructed) when at least one SiPM measured a charge greater than threshold. It is clear that when this condition is satisfied by a coincidence registered in DOWN configuration it cannot be due to a particle downward coming: this is actually a fake event. Once established the charge threshold, it is possible to get the rate of correctly-tagged events from the measurements in UP configuration and the rate of fake events from the corresponding DOWN measurements. These results are reported in Table 2, Table 3, Table 4 and Table 5, columns labeled rate_*UP*_ and rate_*DOWN*_, respectively. In order to compare the measured rates with that of the detector which enables the acquisition, we computed the fraction of external triggers corresponding to correctly tagged events in configuration UP or fake events in configuration DOWN, respectively trig-ext-coin_*UP*_ and trig-ext-coin_*DOWN*_. The right-most column of each table shows the net percentage of external triggers which registered a correctly-tagged event by the Cherenkov prototype, i.e., trig-ext-coin_*UP*_ –trig-ext-coin_*DOWN*_.

The amount of correctly-tagged events in coincidence with an external trigger must be evaluated taking into account the different surface area of the two scintillators with respect to the prototype active surface, the distance between them and that from the bottom scintillator to the Cherenkov detector. For this reason, a Monte Carlo simulation was performed in MATLAB. It starts from points generated uniformly randomly distributed onto two parallel planes, corresponding to the two scintillators. Then, each point of the upper plain is coupled with a point on the bottom plane and the straight line between them is drawn and extended until it reaches a plane below the bottom scintillator plane at a distance equal to that of the Cherenkov prototype. The number of lines that fall into the area corresponding to the Cherenkov-tag prototype with respect to the total amount of lines generated gives the value of the geometric efficiency of the system. In other words, if the efficiency of the Cherenkov prototype matches that of the coincidence between both scintillators, we should expect that the number of correctly-tagged events will be equal to the number of external triggers times the geometric efficiency. Figure 7 and Figure 8 show the trend of the percentage of correctly tagged events with respect to the number of external triggers as a function of SDT set for each acquisition, considering a charge threshold in post-analysis equal to 60 and 70 a.u., respectively. In both plots is reported the value of geometric efficiency as a dashed horizontal line with a strip that accounts for the uncertainty of the simulation.

## 4. Discussion

We reported the plots corresponding to the right-most column just for Table 2 and Table 4, but looking at the values in Table 3 and Table 5 is quite simple to figure out that with a charge threshold equal to 65 a.u. we have something in between Figure 7 and Figure 8, while increasing the charge threshold up to 75 a.u. all values are lower than the expected geometric efficiency. Looking at Figure 7 and Figure 8, it is possible to notice that by choosing a different charge threshold in data analysis makes two distinct acquisitions closer to the expected geometric efficiency. This can be easily explained stating that it is the combination of both acquisition and analysis thresholds which, in the end, determines the effective background noise cut. In support of the previous statement, it is possible to notice that while the percentage of correctly tagged events increases quite linearly with lowering SDT up to 260 a.u., going further down the measured rate begins fluctuating. This is evidence that we are setting a threshold gradually closer to the signal baseline. However, what is important to stress is that it is possible to find a condition in which the percentage of correctly-tagged events with respect to the number of external triggers is very close to the ideal geometric efficiency. This means that the efficiency of the Cherenkov-tag prototype matches that of the detector which triggers the acquisition. Assuming, as usual, that the efficiency of a scintillating tile read-out by a PMT is nearly equal to 100% in detecting cosmic secondary muon events, we can declare successful the proof-of-principle of this new technique for discriminating the incoming direction of these particles.

A direct comparison with the results declared in the feasibility study by means of Geant4 simulations, which, remember, established an efficiency of direction discrimination greater than 98% for muons with energy greater than 1 GeV, is not possible because during the test here described there was not a system to cut the soft component of the secondary cosmic radiation. A specific test with an iron or lead shield above the whole measurement set-up, will be performed as soon as possible. It could be useful also to check the effect of secondary radiation, mainly electrons, generated when a muon crosses a high Z material.

Another significant difference between this work and our previous feasibility study is that here we are not discussing the possibility to get information about the position of the particles crossing the Cherenkov prototype. As it is possible to notice in the example of Figure 6, in the major part of events we have less than 4 SiPMs hit at the same time. In this condition, the fit of charge distribution with a 2D rotated Gaussian function was not feasible. However, we already planned a next test to study a prototype similar to the one discussed here, but with an increased thickness of optical gel. In this way the number of Cherenkov photons produced by the passage of a charged particle will increase and their spreading should be sufficient to reach a higher number of SiPMs. An alternative way to satisfy the requirements in order to be able to reconstruct the muon crossing position consists in detecting a greater number of Cherenkov photons, without increasing their generation. In other words, this means increase the number of SiPMs while keeping constant the thickness of the radiator material, as explored in the feasibility study paper. However, if this solution can be cost-effective for a reduced scale prototype, in sight of the realization of a full scale prototype to couple with the existing MEV telescope ( 1 $m^{2}$ sensitive area), it brings an unwanted increase of complexity and cost. It is possible that we will find that the best solution is a compromise between these two possibilities. In addition, also different SiPMs’ arrangements could be useful to improve the position reconstruction capability of the Cherenkov detector. It is necessary to study them with simulations and, if they give promising results, test them with a bigger prototype. In conclusion, Signal-to-Noise ratio is a key point also for a correct direction discrimination, as it comes from the previous discussion. It is quite evident that increasing the production of Cherenkov photons will help to reach this goal by increasing the signal level. In addition, a different Front-End electronics, based on the MUSIC ASIC [32], could be better suited for the signal level produced. A test board was already designed and we are working also to test this possibility in the next future.

## 5. Conclusions

In this work we discussed the first experimental test of a Cherenkov-tag prototype designed to increase signal-to-noise ratio in muography application by the correct identification of particle incoming direction. This task is usually on a simple geometric criterion or by means of a Time-of-Flight detector which measures the crossing time of the particle at the outermost tracking planes. However, as discussed in our previous paper about the feasibility study of such Cherenkov emission-based technique, time resolution of ToF devices is often not sufficient to distinguish the particle incoming direction. The results here confirm the validity of the Cherenkov-tag detector to correctly solve this issue, even if a solution to improve the signal-to-noise ratio is mandatory in order to test tracking capability of this technique.

## Figures and Tables

**Figure 1 sensors-20-03437-f001:**
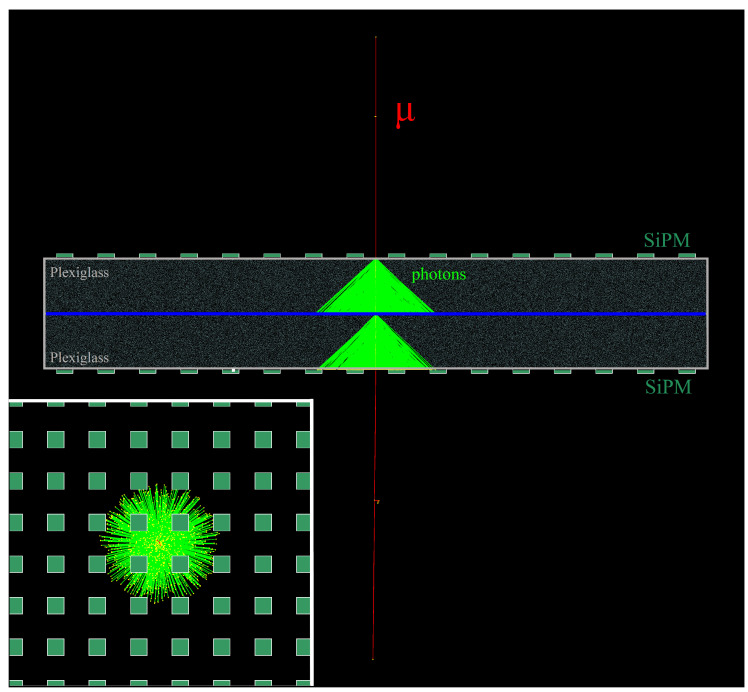
Side view of a muon with kinetic energy $10^{2}$GeV simulated in Geant4. In this simulation, the radiator material is Plexiglass; the size of each tile (biggest horizontal rectangles) is 20×240×240${mm}^{3}$; the external faces are equipped with a regular array of 16×16 silicon photo-multipliers (SiPMs), with size 6×6${mm}^{2}$ (little green rectangles). In the box on the left bottom corner is a front view detail. The yellow dots represent the points at which each Cherenkov (neon green) hits the exit surface of the second radiator.

**Figure 2 sensors-20-03437-f002:**
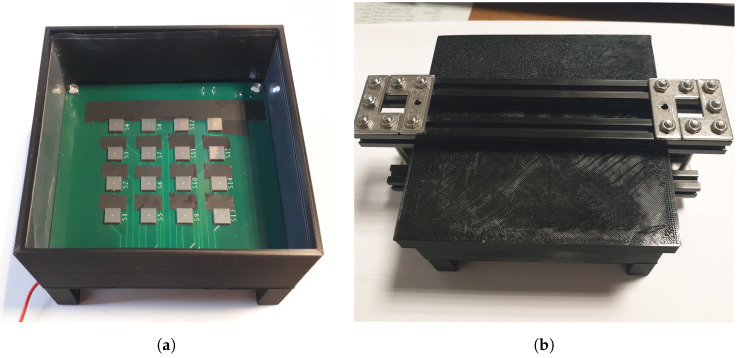
(**a**) View of the opened plastic box with the SiPM board on the bottom and filled with optical gel and (**b**) after the closure with the lid. Board size is $10\times 10{cm}^{2}$ and the regular array of SiPMs, that are 1.5
cm center-to-center equally spaced, is centered on the board. SiPMs have $6\times 6{mm}^{2}$ nominal surface area.

**Figure 3 sensors-20-03437-f003:**
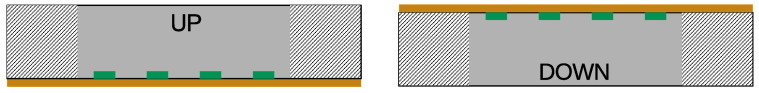
2D sketch of the two measurement configurations required to measure muon flux rate detected by the Cherenkov-tag prototype and the corresponding fake events rate. The patterned areas correspond to the optical gel insensitive volume.

**Figure 4 sensors-20-03437-f004:**
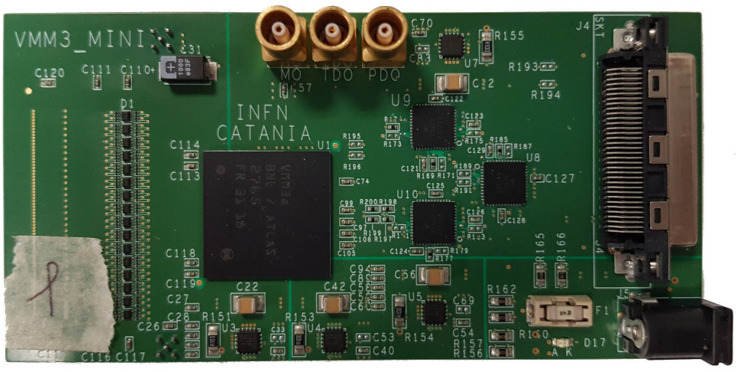
Front-End board with VMM3a chip installed.

**Figure 5 sensors-20-03437-f005:**
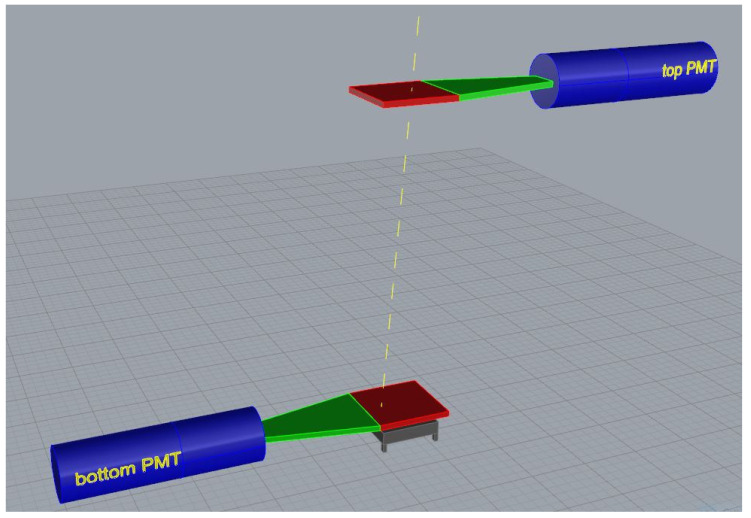
3D sketch, not to scale, of the experimental setup for the test of Cherenkov-tag prototype. In order to more easily distinguish each component of the trigger system, they are differently colored: Photo-Multiplier tubes (PMTs) are blue, light guides are green and the scintillating tiles are red.

**Figure 6 sensors-20-03437-f006:**
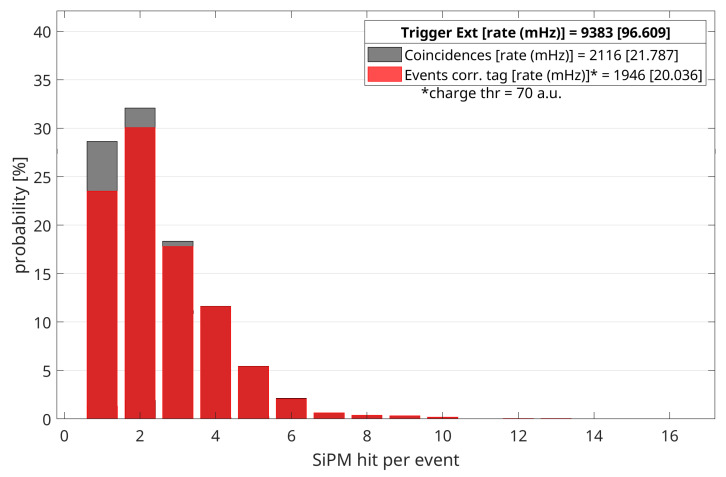
Bar plot of the probability of hitting a certain number of SiPM for the acquisition in UP configuration and SDT = 254. Red and grey bars refers to the probability that *n* SiPMs are stricken with a measured charge greater than threshold, equal to 70 a.u. in this example.

**Figure 7 sensors-20-03437-f007:**
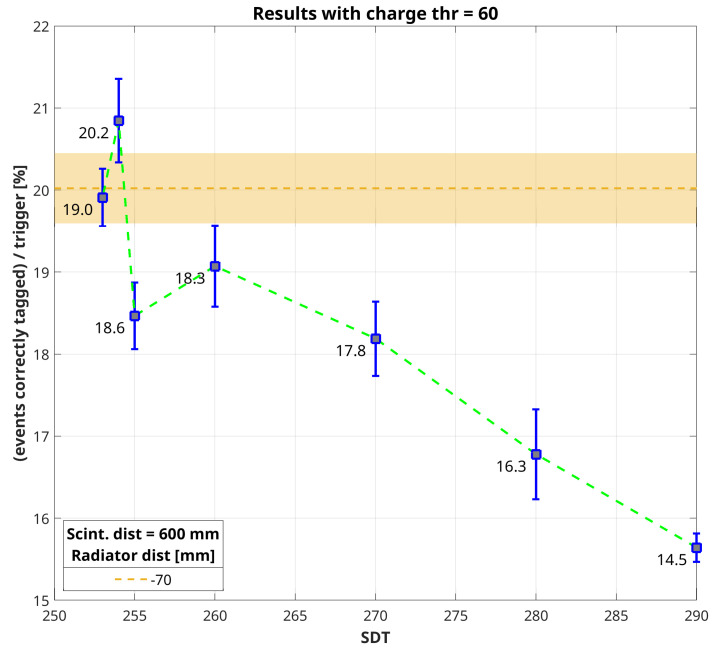
Comparison between the percentage of correctly tagged events with respect to the number of external triggers and the geometric efficiency calculated with a Monte Carlo simulation (dark yellow horizontal dashed line). This plot refers to values reported in Table 2, with charge threshold equal to 60 a.u. Values reported in the plot near the squared markers correspond to the net rate of correctly-tagged events, i.e., rate_*UP*_–rate_*DOWN*_ in Table 2.

**Figure 8 sensors-20-03437-f008:**
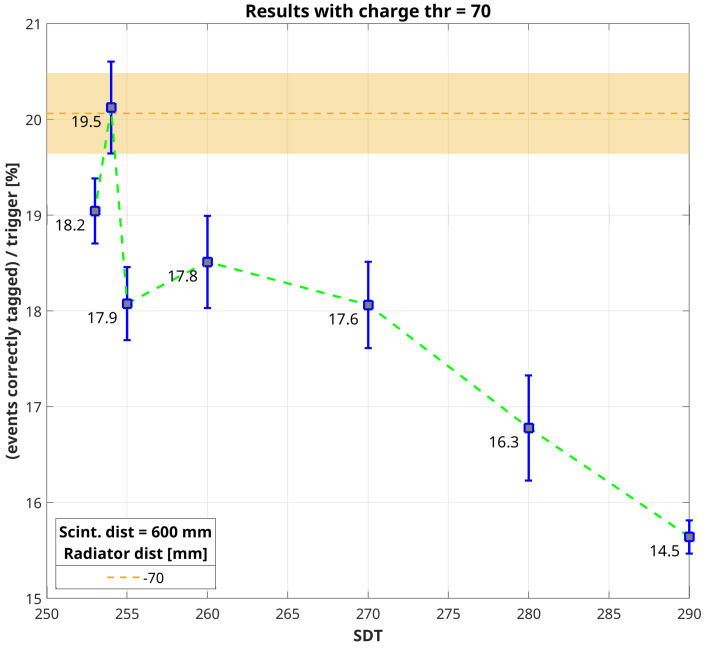
Comparison between the percentage of correctly tagged events with respect to the number of external triggers and the geometric efficiency calculated with a Monte Carlo simulation (dark yellow horizontal dashed line). This plot refers to values reported in Table 4, with charge threshold equal to 70 a.u. Values reported in the plot near the squared markers correspond to the net rate of correctly-tagged events, i.e., rate_*UP*_–rate_*DOWN*_ in Table 4.

**Table 1 sensors-20-03437-t001:** VMM chip SDT (common threshold for all channels) values set for test measurements with Cherenkov-tag prototype and corresponding acquisition time for both configurations, UP and DOWN.

SDT [a.u.]	$t_{\mathrm{UP}}$ [s]	$t_{\mathrm{DOWN}}$ [s]
290	586,911	141,961
280	59,084	96,248
270	97,529	94,768
260	87,791	96,634
255	149,435	139,809
254	97,123	67,795
253	184,055	324,834

**Table 2 sensors-20-03437-t002:** Measured rate of correctly-tagged events (rate_*UP*_) [fake events, (rate_*DOWN*_)] and corresponding amounts of external trigger with an event measured by Cherenkov prototype also (trig-ext-coin). Last column shows the net percentage of external triggers which correspond to correctly-tagged events, with a charge threshold equal to 60 a.u.

SDT	Rate_*UP*_	trig-ext-coin_*UP*_	Rate_*DOWN*_	trig-ext-coin_*DOWN*_	Correctly-tag
[a.u.]	[mHz]	[%]	[mHz]	[%]	[%]
290	14.7±0.2	15.9±0.2	0.27±0.04	0.29±0.05	15.6±0.2
280	16.5±0.5	17.0±0.5	0.25±0.05	0.27±0.05	16.8±0.5
270	18.3±0.4	18.8±0.4	0.58±0.08	0.62±0.08	18.2±0.5
260	19.0±0.5	19.8±0.5	0.63±0.08	0.68±0.09	19.1±0.5
255	19.6±0.4	20.3±0.4	0.94±0.08	1.79±0.16	18.5±0.4
254	21.6±0.5	22.3±0.5	1.33±0.14	1.42±0.15	20.8±0.5
253	20.0±0.3	21.0±0.3	1.05±0.06	1.10±0.06	19.9±0.4

**Table 3 sensors-20-03437-t003:** As in Table 2, for a charge threshold equal to 65 a.u.

SDT	Rate_*UP*_	trig-ext-coin_*UP*_	Rate_*DOWN*_	trig-ext-coin_*DOWN*_	Correctly-tag
[a.u.]	[mHz]	[%]	[mHz]	[%]	[%]
290	14.7±0.2	15.9±0.2	0.27±0.04	0.29±0.05	15.6±0.2
280	16.5±0.5	17.0±0.5	0.25±0.05	0.27±0.05	16.8±0.5
270	18.3±0.4	18.8±0.4	0.58±0.08	0.62±0.08	18.2±0.5
260	18.7±0.5	19.5±0.5	0.63±0.08	0.68±0.09	18.8±0.5
255	19.2±0.4	19.8±0.4	0.68±0.07	1.30±0.13	18.5±0.4
254	20.8±0.5	21.6±0.5	0.86±0.11	0.91±0.12	20.7±0.5
253	19.5±0.3	20.5±0.3	0.83±0.05	0.86±0.05	19.6±0.3

**Table 4 sensors-20-03437-t004:** As in Table 2, for a charge threshold equal to 70 a.u.

SDT	Rate_*UP*_	trig-ext-coin_*UP*_	Rate_*DOWN*_	trig-ext-coin_*DOWN*_	Correctly-tag
[a.u.]	[mHz]	[%]	[mHz]	[%]	[%]
290	14.7±0.2	15.9±0.2	0.27±0.04	0.29±0.04	15.6±0.2
280	16.5±0.5	17.0±0.5	0.25±0.05	0.27±0.05	16.8±0.5
270	18.2±0.4	18.7±0.4	0.58±0.08	0.62±0.08	18.1±0.5
260	18.2±0.5	19.0±0.5	0.43±0.07	0.47±0.07	18.5±0.5
255	18.5±0.4	19.1±0.4	0.52±0.06	1.00±0.12	18.1±0.4
254	20.0±0.5	20.7±0.5	0.58±0.09	0.61±0.10	20.1±0.5
253	18.8±0.3	19.7±0.3	0.66±0.05	0.69±0.05	19.0±0.3

**Table 5 sensors-20-03437-t005:** As in Table 2, for a charge threshold equal to 75 a.u.

SDT	Rate_*UP*_	trig-ext-coin_*UP*_	Rate_*DOWN*_	trig-ext-coin_*DOWN*_	Correctly-tag
[a.u.]	[mHz]	[%]	[mHz]	[%]	[%]
290	14.7±0.2	15.9±0.2	0.27±0.04	0.29±0.05	15.6±0.2
280	16.5±0.5	17.0±0.5	0.25±0.05	0.27±0.05	16.8±0.5
270	18.0±0.4	18.5±0.4	0.58±0.08	0.62±0.08	17.8±0.4
260	17.5±0.4	18.2±0.5	0.34±0.06	0.38±0.06	17.9±0.5
255	17.7±0.3	18.3±0.4	0.39±0.05	0.74±0.10	17.6±0.4
254	19.3±0.4	19.9±0.5	0.49±0.08	0.52±0.09	19.4±0.5
253	18.2±0.3	19.0±0.3	0.51±0.04	0.53±0.04	18.5±0.3

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
