# Peer review of "Proof-of-Principle of a Cherenkov-Tag Detector Prototype"

_sensors, 2020, doi:10.3390/s20123437_

Round 1

Reviewer 1 Report

General comment:

This is very interesting study to establish a new technique utilising a direction of Cherenkov light in order to solve the mis-acquisition of muon-radiography. The idea is very simple, and the study procedure looks well-considered, Very Interesting !
Unfortunately, the manuscript is diffused a bit, for example, Section 1 (Introduction) seems a textbook of muon-telescope, Section 2.2 (SiPMs FE) seems a technical description about VMM chip. Your manuscript should concentrate on the main research, i.e. it is better to be simpler by omitting an unnecessary description.

Detailed comments;

Abstract
"VMM3a" is referred without any description.
It's not necessary to refer a chip name here, just a "front-end electronics for SiPM signals" is enough as an abstract.

Section 2.1

  • Line 116: "mbar" should be converted to "Pa" (bar is not SI).
  • Figure 2, it's better to put scale upon this figure, although the dimension of prototype is described in a body.

Section 2.3

  • Why the prototype is placed just behind the bottom scintillator ? I mean, putting prototype just over the bottom scintillator looks natural...
  • Line 160: "LVCMOS" is used without any description.
    I can understand LVCMOS without a full text, but I'm not sure this is common for every reader of this journal... Thus it's better to put this abbreviation with a full text here.
  • Line 174: "preliminary calibration", What kind of calibration is performed here ?
  • Line 175: Again, "SD" is used without a full text.

Section 3

  • Line 179: "Our experience suggests" is not a scientific description. It's better to put more scientific description, or just to say that gain and peak time was set to be minimum...
  • Line 182: "So,...". "So" is too casual, not suitable as an academic article. You should use "Therefore" or "Thus" instead.

Again, Section 1, 2.2 and 2.3 is too diffused.
It's better to keep your manuscript as short as possible to be more readable. Your study itself is very interesting, so please consider to make it more readable.
And I'm looking forward to reading your next result with a thicker radiator.

Author Response

Dear Sir,

Thank you for your detailed and useful review, your comments gave a great chance to improve the quality of the manuscript. We really appreciate your interest in our work and its future developments. 

Following your tips, the first part of the introduction was shortened. However, the broad context in which our work is placed continues to be described and all references are preserved in order to give a quick access to an exhaustive catalogue of papers about muography for the interested reader. This is required in the journal guidelines.

Also the VMM chip description was shrinked, avoiding all details that can be found in ref [29].

Dimensions of the prototype are now reported in the caption of Figure 2. We preferred to not directly modify the picture, but the information you mentioned is now quickly accessible. 

There is not a “scientific” reason because the prototype was placed just below the bottom scintillator, but, simply, the mechanical supports of scintillators and that of the Cherenkov prototype make this arrangement the easiest to build. We searched for a placement of both scintillating tiles and prototype in order to select muons as much as possible near to the vertical direction, without reducing too much the rate. This allows to maximise the number of particles that pass through the sensitive volume of the prototype with respect to the borders not covered by SiPMs. Just for curiosity after reading your comment, we performed a couple of simulations in order to evaluate the zenith angular acceptance of the prototype with respect to that of the telescope constituted by the two scintillating tiles. You can find the results in the attacched file.

You are completely right about the previous mention of “SD” at the end of section 2.3. Indeed, That sentence about preliminary calibration of the VMM was repeated 9 lines after the beginning of section 3, which is now the only correct placement. It was a copy-and-paste mistake. The preliminary calibration mentioned in the manuscript consists in a fine tuning of the threshold of each VMM channel, with respect to the common threshold, in order to equalize as much as possible the dark current rate. It was performed by means of an oscilloscope.

Other minor revisions you requested have been included in the manuscript and the changes applied is highlighted in yellow.

Reviewer 2 Report

The paper is well written. The authors describe the hardware and the experiments to prove that the system is working properly. While the data prove the concept will work, it will be more useful when the full-up system is built and functional. This is an important step to show that the individual parts of the prototype will work together.

Line 151 – instead of Figure 4 I am pretty sure it should read Figure 5.

Author Response

Dear Sir,

We are very grateful to you for the endorsement of our manuscript. The realization and testing of a full-scale prototype to fit the dimensions of the MEV telescope (1m2) are in our plans, but, unfortunately, the COVID-19 outbreak delayed it to an unpredictable future. We hope that you could follow our next results as they will be delivered. 

You are certainly right, the reference at line 151 is to Figure 5. It was adjusted in the text.

Reviewer 3 Report

It's with pleasure that I read the manuscript. 

The paper is well written and the procedure is easy to follow. I appreciated the Monte Carlo simulation to quantify the detector's accuracy as a function of the common threshold.

I think that the results and discussion sections are complete, however I would recommend the authors amplify further on whether a different number or arrangement of the photodetectors is necessary in addition to the increased thickness of the optical gel.

Also, in the conclusion, could the authors briefly outline a possible strategy to increase SNR?

Author Response

Dear Sir,

We are very grateful to you for the endorsement of our work. 

We expanded the Discussion section talking about the topics mentioned in your comments, i.e. the SiPM number increase and arrangement and the possible solutions to improve the SNR. You will find it highlighted in the manuscript at the end of Section 4.